# Identification of Organic Volatile Markers Associated with Aroma during Maturation of Strawberry Fruits

**DOI:** 10.3390/molecules26020504

**Published:** 2021-01-19

**Authors:** Samuel Macario Padilla-Jiménez, María Valentina Angoa-Pérez, Hortencia Gabriela Mena-Violante, Guadalupe Oyoque-Salcedo, José Luis Montañez-Soto, Ernesto Oregel-Zamudio

**Affiliations:** 1Instituto Politécnico Nacional, Centro Interdisciplinario de Investigación para el Desarrollo Integral Regional (CIIDIR), Unidad Michoacán, Justo Sierra 28, Col. Centro, Jiquilpan 59510, Mexico; samimac93@hotmail.com (S.M.P.-J.); vangoa@ipn.mx (M.V.A.-P.); hmena@ipn.mx (H.G.M.-V.); goyoque@ipn.mx (G.O.-S.); montasoto@yahoo.com.mx (J.L.M.-S.); 2Instituto Politécnico Nacional, Programa de Doctorado en Ciencias en Bioprocesos, Unidad Profesional Interdisciplinaria de Biotecnología (UPIBI), Av. Acueducto, Barrio la Laguna Ticoman, Ciudad de México 07340, Mexico

**Keywords:** aroma, headspace solid-phase microextraction, gas chromatography-mass spectrometry, volatile organic compounds

## Abstract

In the present study, organic volatile markers of three strawberry varieties (Albion, Festival and Frontera) during the maturation process were investigated. Forty metabolites associated with aroma in fresh strawberries were monitored during seven stages of maturation using gas chromatography–mass spectrometry (GC-MS) equipped with headspace-solid phase microextraction (HS-SPME). The data were evaluated using multivariate analysis to observe correlations between the organic volatile compound profile and the seven phenological stages of maturation for each strawberry variety. The dynamic levels of butanoic acid methyl ester, hexanoic acid methyl ester, octylcyclohexane, cyclohexane,1,1,2-trimethyl, linalool, tetradecane, and α-muurolene underwent distinctive changes in concentration during the maturation process. The multivariate analysis also allowed the identification of these compounds as possible volatile markers to measure the maturation of strawberry fruits in all three varieties. These findings highlight the importance of the timing of harvest and maturation stage in each variety to preserve or improve the desirable aromatic characteristics of strawberry fruits.

## 1. Introduction

Volatile organic compounds (VOCs) are known to be responsible for aroma. In strawberry fruits, aroma is an organoleptic characteristic that directly influences consumption. Olfaction is both chemically sensitive in itself and closely linked to taste, making aroma, and the vast array of chemical reactions responsible for it, a critical criterion for consumer preference and important marker of quality in fruit [1].

Aroma plays an important role in strawberry cultivation during all stages of development, including an attractant of beneficial insects and direct defense against pests [2], as a warning response against pathogen attack [3], and in plant-to-plant volatile communication [4]. Thus, we can infer that fruits’ aroma is the result of their evolutionary history, resulting from constant interaction with the surrounding environment [5].

Strawberries are produced commercially in 76 countries, the most producing nations are USA, Mexico, Turkey, and Spain. In Mexico the most cultivated varieties of strawberry fruits are Albion Festival and Frontera, since they are the most adapted and with the highest production in this country [1]. Strawberry fruit is a clear example of how aroma can be a highly complex characteristic since around 360 VOCs have been reported in relation to aroma [6]. Different strawberry varieties possess distinctive VOCs, which collectively result in a specific compound profile for each variety [7]. For some fruits, such as apples (*Malus domestica*) it is possible to identify different volatile profiles from peels [8]. Thus, aroma can be a fingerprint to distinguish among varieties, depending on the concentration, diversity, and perception threshold of the VOCs present in the fruit [9]. In addition to variety, the degree of maturation of the fruit has an effect on its VOC fingerprint [10]. VOC profiles have been shown to increment over the course of maturation in climacteric fruits, and a similar effect could occur in non-climacteric fruits also [11].

Strawberry fruits are a non-climacteric fruit, which means that once they are cut, the production of hormones will decrease, which will lead to changes in the biosynthesis of secondary metabolites including VOCs, leading ultimately to the production of distinct aromatic notes; at the same time, the process of senescence will begin, and therefore the organoleptic characteristics like color and texture will change [12]. The senescence could lead to defects in fruit quality if fruits are not harvested properly [13]. There are few studies demonstrating how strawberry fruit maturation impacts VOC profile, and how this profile of strawberry fruits is highly sensitive to fruit harvest. These studies show that esters are the volatile compounds most correlated with strawberry fruit maturation [14,15]. Monitoring the VOC profile during the maturation process could provide specific markers of the optimal time to harvest, guaranteeing a standardized aroma to consumers, and ensuring the highest possible quality. VOC profiles have been analyzed for a variety of mature strawberry fruits, showing differences in the concentration and diversity of compounds. However, the behavior of the VOC profile over the process of fruit maturation has not been explored. In addition, it is unknown how VOC changes during maturation differ among varieties that present distinctive profiles and unique VOCs. As such, the objective of this work was to identify possible volatile organic compounds related to aroma to monitor the maturation of strawberry fruits in order to generate reference markers to optimize cultivation and harvest better quality product.

## 2. Results and Discussion

Among all of the maturation stages in the three varieties, a variety of VOCs composed of 42.5% esters, 25% alkanes, 20% terpenes, 7.5% alcohols, 2.5% aldehydes, and 2.5% acids were identified. These results are consistent with those of El Hadi, Zhang, Wu, Zhou and Tao [1] where the most representative chemical families of strawberry fruits are esters and terpenes. Looking at the results in a more detailed way shows patterns of increase and decrease of different VOCs over the course of the seven maturation stages analyzed for each variety.

In the Albion variety, α-muurolene, octocyclohexane, and tetradecane had significantly higher concentrations during the first maturation stages (AL0, AL1 and AL2), then butanoic acid methyl ester increased in abundance in stages AL3–AL6 (Table 1). Linalool also increased in concentration with increasing fruit maturity. This corresponds with previous reports, which clearly state that this VOC is characteristic of mature fresh strawberry fruits [16]. On the other hand, α-muurolene was at its maximum level during stage AL0 and decreased in concentration with increasing maturation. Decreasing α-muurolene as fruits mature has been reported for avocado *Persea Americana* [17] Kirillov, et al. [18] detected α-muurolene as one of the key VOCs of fruits of green strawberry *Fragaria viridis* Weston. Butanoic acid methyl ester, which reached its highest concentration during stages AL3–AL6, has been reported as a key VOC in this variety [19,20], and there are studies that define butanoic acid methyl ester as highly important in the characteristic aroma of other varieties of strawberries such as var. Fortuna and var. Camarosa [21,22]. One implication of this series of results in Albion variety is that it opens up the possibility of using the concentration of VOCs such as linalool, α-muurolene, and butanoic acid methyl ester as indicators to measure the maturity level of fresh strawberry fruits, offering a cue of optimal timing of harvest. However, we must be cautious, since these findings may not be transferable to other varieties.

In the Festival variety, tetradecane was the VOC with the highest concentration during stages FE1 and FE2, the compound with the highest concentration was nonadecane, and in stages FE3–FE6, the compound with the highest concentration was hexanoic acid methyl ester (Table 2). Butanoic acid methyl ester increased in concentration during the maturation of the strawberry fruits and was the compound with the highest concentration during those stages, which suggests that this a key compound that could be used as an indicator of maturity in this variety. It has been reported to be the main compound responsible for the fruity aroma of many strawberry varieties, participating actively in the aromatic profiles of four strawberry varieties: Camarosa, Crystal, Monterey, and Portola [23].

On the other hand, 3-hexen-1-ol, acetate, (*E*)- which is also an esterified compound, presented its maximum concentration in stage FE3. This shows that not all compounds from the ester family present high concentrations in mature stages. This is interesting since most studies report that these esters are found in high concentrations in mature fruits [11], but our results show that not all esters have the same behavior, and only some may be useful as indicators of complete maturity.

The alkane family may also provide information on the maturity of strawberry fruits of the Festival variety. Tetradecane was the most abundant compound in the Festival variety at the FE0 stage and Nonadecane was the most abundant compound in stages FE1 and FE2. This is in accordance with observations from the work reported by Nan, et al. [24] describing the group of alkanes that impact the aroma of certain strawberries in initial stages of maturation. Hexadecane reaches its highest concentration during the FE1 stage, making it another key compound in this stage of maturity, and its presence in high quantities indicates immature fruits. It has been reported that these compounds of the alkane family are responsible for herbaceous notes characteristic of fruits that have not finished developing [21]. The implications of having identified possible volatile markers of specific stages of maturation are quite promising, since they allow them to be a reference point for the agronomic management for cultivation.

For the Frontera variety, the VOC with the highest concentration in the FR0 stage was cyclohexane, 1,1,2-trimethyl. The presence of this compound has been reported in other strawberries at initial maturation stages when 100% of the fruit surface is green and is related to an herbaceous aroma [25]. In FR1, the most abundant compound was methyl acetate, which has been identified in varieties of wild strawberry and gives a floral note to those fruits [26]. In the FR3–FR5 stages, the most abundant compound was most abundant was 1-butanol, 2-methyl-, acetate; this compound has an impact on the aroma of strawberry fruits in late stages of maturation and has been previously identified in other strawberry varieties such as Camarosa [27]. Finally, in stage FR6, hexanoic acid methyl ester was the VOC with the highest concentration, which indicates that this compound is closely related to the aroma of mature strawberries (Table 3). 

In the Frontera variety, the VOCs that define the earliest maturation stages were terpenes. Surprisingly the majority of these reached their maximum concentration when the fruit’s surface was still 100% green. This could explain the noticeable floral aroma at this stage of maturation in this variety. These terpenes are known to be responsible for floral aromatic notes, which are not characteristic of mature fruits, but rather participate in some aspects of initial development, such as attractants for beneficial insects [28]. This information could provide an opportunity to carry out more effective pollination techniques in greenhouses by using pollinators during the stages with the highest production of floral aromas emitted by terpenes. One example is the work by Ceuppens, et al. [29] where the Sonata variety was found to produce a higher pollination index than the Elsanta variety; this was attributed to the higher concentration of floral aromas emitted by the Sonata variety. Some compounds from the ester family increased as fruits matured. For example, isopropyl butyrate reached its maximum concentration during the last stage of maturation in Frontera variety fruits. Ester family contribute to the characteristic aroma of mature strawberry fruits, since these compounds are responsible for the fruity aromatic notes [16].

The VOCs profile of the stages of maturation can be classified according to the concentrations of the volatiles identified as shown in Figure 1. A heatmap provides intuitive visualization of data. Each colored cell on the map corresponds to a concentration relative value, with metabolites in rows and maturation stage in columns. Hierarchical clustering is done of both the rows and the columns. The columns/rows were re-ordered according to the hierarchical clustering result, putting similar observations close to each other. Visualizing the maturation stage in this way can help to find the variables that appear to be characteristic for each sample cluster. 

It can be seen that stages AL4 and AL5 are grouped on the same classification branch, and therefore their VOC profiles are not significantly different; on the other hand, AL6 is not paired with any other stage, so the AL6 variety/stage is different from the rest in terms of its VOC profile (Figure 1a). This could indicate that the aroma of this variety does not completely finish developing until the fruit reaches 43–44 DAA. This classification can be explained based on the fact that the compounds with the strongest positive correlation in the AL6 stage are esterified compounds. It has been reported that this chemical group is one of the most influenced by fruit maturation, and esterified VOCs are considered to be characteristic of mature fruits and are responsible for fruity aromas [30].

At the same time Figure 1a, shows valuable information for each stage of maturation that could be used to optimize cultivation and harvest. An example is the use of nutrients deployed to the crop during fertilization. Sangha and Agehara [31] discovered that fertilization of strawberry fruits with nitrogen of the Florida radiance variety is effective only up to 30 DAA, and using the same amount of nitrogen after that point had no effect on strawberry fruit production. Using the VOC profile in Figure 1a could help to identify the moment when applying nitrogen will no longer have a significant effect on the crop; this tool shows that while stages AL2 and AL3 present the same coloration, they do not share the same VOC profile, and the AL3 profile belongs to the 30–35 DAA time frame, which would provide a cue of when to decrease the amount of fertilizer. In the case of stages AL4 (41–42 DAA) and AL5 (43–44 DAA), the surface color of the fruit is no longer the same since AL4 has 30% of the surface red and AL5 has 60% of the surface red. However, their VOC profiles do not differ significantly, so these stages could be used as export quality fruit since according to Parvez and Wani [32] the quality of fruit for export only takes into account the fruit color, some companies have opted for treatment with modified atmospheres to finish the development of red coloration during the shipping time of strawberry fruits, leaving aside organoleptic characteristics like aroma. In the case of the Albion variety, the fruits could be harvested in stage AL4 and their profile would not differ significantly with respect to AL5, giving producers a wider harvest margin.

Interestingly, having worked with different varieties allowed us to observe that stages FE5 (41–42 DAA) and FE6 (43–44 DAA) are found on the same branch of the classification tree in Figure 1b, which indicates that there is no significant difference between these stages in aroma-related VOC profile. This shows that in the Festival variety, the VOC profile can reach its final state before fruits reach 100% red surface color. This contrasts with the Albion variety, in which the aromatic profile is not fully developed until fruits are in the last maturation stage; thus, fruit variety is a determining factor when it comes to maturity [33]. One way to utilize the information in 1b as a tool to optimize strawberry cultivation in greenhouses is the adequate timing of pest management measures. Work by Hori, et al. [34] reveals that strawberry fruits are attacked by the pest *Galerucella* spp. when fruits present maximum herbaceous aroma, which is a characteristic of stages FE0 and FE1. The VOC profile of the Festival variety could be used to develop a more adequate pest control program as a function of the stage during which the pests are most attracted.

In the case of the stages of maturation in the Frontera variety, this is similar to the Albion variety. In Figure 1c, it is evident that the last stage of maturation (FR6) is alone on the classification tree, which indicates that in this variety, the VOCs related to aroma reach their maximum when the fruit surface is 100% red and 43–44 days have passed since anthesis; at the same time, the chemical family with the highest positive correlation is the esterified COVs, which as mentioned above, are the most representative of mature fruits [1].

To distinguish the most important metabolites for each variety during the maturation process, variable importance in the projection (VIP) graph was used. VIP is a weighted sum of the squares, it takes into account to count the amount of variation Y-explained. Higher VIP scores show the metabolites that had a higher level of importance within each variety. The generated VIP graph classified the individual metabolites by their ability to separate the maturation stages in each variety, respectively.

In Figure 2a, it can be observed that for the latest maturation stages of the Albion variety, the compound with the strongest positive correlation was hexanoic acid methyl ester, which belongs to the chemical family of the esters. This family is one of the most influential in terms of the characteristic aroma of strawberries, so it can be inferred that this compound can be used as a volatile marker for determining the optimal maturation stage for the Albion variety [35]. On the other hand, hexanoic acid methyl ester showed a negative correlation with respect to fruit in initial stages, and the compound that was most representative of these stages of maturation was octylcyclohexane, which belongs to the family of the alkanes, which are closely associated with the herbaceous aromas of immature fruits [24].

For the Festival variety, hexanoic acid methyl ester was the most representative compound for the latest stages of maturation (FE4, FE5, and FE6), making it potentially useful as an indicator of maturity for this variety, while heptylcyclohexane was the most representative of the initial stages (FE0 and FE1). This is similar to the report by Du and Rouseff [21] that hexanoic acid methyl ester is one of the main VOCs involved in the aroma of mature strawberry, and therefore, that hexanoic acid methyl ester can be used as a volatile marker to identify the degree of maturity in strawberry fruits of the Festival variety (Figure 2b).

In the case of the Frontera variety, the compound most strongly associated with the latest stages of maturation (FR4, FR5, and FR6) was butanoic acid methyl ester, which belongs to the ester family and, like in the previous cases, is characteristic of mature fruit, it has been mostly reported in strawberry and is characterized by the fruity aromatic note and therefore this compound could be an indicator of optima maturation in the Frontera variety [20] (Figure 2c).

Taken together, the univariate and multivariate statistical analyzes show that, despite the significant variability between the volatile profile at each stage of maturation, a proportion of the metabolic variability between strawberry varieties is conserved from one variety to another, as determined by established based on metabolite profiles. The reason for this is not clear, but it may have something to do with the particular characteristics that each variety presents. The most interesting fact is that evidence was found that there are differences in the maturation process between strawberry varieties, examining their volatility profile at each stage of maturation. These differences may be appropriate for tracking specific strawberry varieties. This is an important issue for future research including other fruits.

Figure 3 shows a diagram of the possible metabolic pathways involved in the synthesis of the nine main VOCs obtained from the analysis in Figure 3. These nine VOCs belong to three chemical families: esters, terpenes and alkanes. The ester chemical family results from the lipoxygenase metabolic pathway, which involves the metabolism of fatty acids; this allows us to deduce that esters are secondary metabolites derived from the methyl esterification of some fatty acids [9]. We speculate that the metabolites from the terpene chemical family are derived from the catalysis of some phenolic compounds from the biosynthesis in the mevalonic acid pathway. This explains why terpenes have been cataloged as compounds with antioxidant activity [36]. Finally, the alkane group compounds are the endpoint of the hydrolysis of some fatty acids. It is worth pointing out that these have been classified as characteristic compounds of herbaceous aromas and are considered key compounds in immature strawberry fruits [37]. The metabolic pathways for volatile biosynthesis, including esters, terpenes, alkanes, amino acids, fatty acids, and carotenoids, are diverse and often highly integrated with other parts of the primary and secondary metabolism. These results therefore need to be interpreted with caution. More research is needed to identify the key substrates and enzymes involved in order to point to a more extensive explanation of the possible metabolic pathways during fruit maturation. However, this study helps us to better understand the maturation process of non-climacteric fruits. It also shows evidence of an effect of variety on aroma development. The times of harvest and the ripening stage are very important in each variety to preserve or enhance the desirable aromatic qualities of strawberry fruits. This work lays the foundations for future studies related to the harvest of non-climacteric fruits and studies to optimize the application of agronomic techniques to increase the quality of strawberry fruits. Additionally, the information presented could influence the planning and management of strawberry crops, including integrated pest and disease control.

## 3. Materials and Methods 

### 3.1. Plant Material

Strawberry fruits (*Fragaria × ananassa*) Albion variety, Festival variety, and Frontera variety were collected at seven stages of maturation from an experimental field located in the Jacona municipality of Michoacán, Mexico (19.961566° N, 102.317995° W). All of the strawberry varieties were cultivated in a macrotunnel under the same fertilization, irrigation and agronomic management conditions. Harvest and classification of fruits was in accordance with Ménager, Jost and Aubert [14] and Parra-Palma, Úbeda, Gil, Ramos, Castro and Morales-Quintana [16]. Seven maturation stages based on the number of days after anthesis (DAA) and the percentage of red color on the surface of each of the harvested fruits were defined, leading to the following criteria for each stage: stage 0: 20–25 DAA, 0% red; stage 1: 26–30 DAA, 15% red; stage 2: 30–35 DAA, 20% red; stage 3: 36–38 DAA, 30% red; stage 4: 39–40 DAA, 60% red; stage 5: 41–42 DAA, 80% red; and stage 6: 43–44 DAA and 100% red. The stages of maturation were combined with the variety to name each maturation stage-variety combination examined: Festival variety: FE0, FE1, FE2, FE3, FE4, FE5, and FE6; Frontera variety: FR0, FR1, FR2, FR3, FR4, FR5, and FR6; Albion variety: AL0, AL1, AL2, AL3, AL4, AL5, and AL6. After harvesting, fruits were transported immediately to the laboratory, maintaining a temperature of 4 ± 1 °C, and they were stored in refrigeration for no more than 24 h before analysis. Prior to each analysis, it was verified that the fruits were free of physical or biological damage.

### 3.2. Chemical Reagents

All reagents were purchased from Sigma-Aldrich, St. Louis, MO, USA. The volatile standards were methyl acetate (≥99%), methyl isovalerate (≥99%), isopropyl butyrate (≥99%), hexanoic acid methyl ester (≥99%), 3-carene (≥99%), limonene (≥99%), 1-hexanol, 2-ethyl (≥99%), linalool (≥99%), nonanal (≥99%), nonadecane (≥99%), tetradecane (≥99%), α-farnesene (≥99%), α-muurolene (≥99%), and ethyl *n*-decanoate (≥99%). Sodium chloride (NaCl) (99.5%).

### 3.3. Preparation of Samples for Volatile Analysis

For VOC analysis, the samples were prepared according to Lu, et al. [38] with modifications, used 100 g of fresh strawberry from each respective maturation stage were used, these fruits were homogenized with 100 mL of a 20% NaCl solution (*w*/*v*) the samples were frozen in liquid nitrogen, and stored at −10 °C until analysis, the samples were analyzed in triplicate.

### 3.4. Headspace-Solid Phase Microextraction (HS-SPME)

Headspace solid phase microextraction (HS-SPME) was used to analyze the VOCs in strawberry samples. The test was carried out using a 50/30 μm capillary fiber (DVB/CAR/PDMS, Supelco, Bellefonte, PA, USA), which was protected by a manual holder (57330-u, Supelco, Bellefonte, PA, USA). Prior to analysis, each sample had an equilibrium time of 30 min in the acrylic reactor (30 ± 1 °C). The analyzes were carried out according to Camelo, et al. [39] with modifications, the fiber was introduced into the acrylic reactor with the sample, the fiber was exposed for 3 min to the head space in the acrylic reactor, which was at a temperature of 30 ± 1 °C. Once the exposure time had passed, the fiber was placed into the holder and transported to the injector of the gas chromatograph mass spectrometer (GC/MS), which had a temperature of 230 ± 1 °C, the fiber was introduced into the injector and exposed for 3 min to desorb the VOCs from the strawberry samples, all samples were analyzed whit three technical replicates. Conditions of temperature and relative humidity in the laboratory analysis were 20 ± 3 °C and 67 ± 5%, respectively.

### 3.5. Gas Chromatography Mass Spectrometry (GC/MS)

A gas chromatograph (Clarus 680, Perkin-Elmer Inc., Walham, MA, USA) coupled with a mass spectrometer (Clarus SQ8T, Perkin-Elmer Inc., Walham, MA, USA) equipped with a phase capillary column: 5% diphenyl, 95% dimethylpolysiloxane, 30 m in length, 0.32 mm in ID, 0.25 μm of film thickness (Elite-5 MS, Perkin-Elmer Inc., Walham, MA, USA) was used. The temperature of the injector was 230 ± 1 °C. A constant flow of helium gas at 1 mL/min with an initial wait time of 0.05 min was used. The temperature program for the oven was: 30 ± 1 °C for 2 min, which was then increased to 140 °C at 9 ± 1 °C/min, remaining at this temperature for an additional 5 min. An electron impact ionization source (70 eV) was used in exploration mode. The temperatures of the transfer line and the ionization source were 230 and 250 ± 1 °C, respectively.

### 3.6. Identification and Quantification of Volatile Organic Compounds Related to Strawberry Aroma

VOCs were identified by comparing the mass spectrum of each peak found in the simple chromatographs with mass spectra from the database of the National Institute of Standards and Technology (NIST/EPA/NIH Mass Spectral Library version 2.2. 2014). Once the compounds were identified, the Kovats Index (KI) was calculated, and the quantification for each was carried out using internal standard. For identification of VOCs, the C7–C40 saturated alkane standards (Certified reference material, Sigma Aldrich, St. Louis, MO, USA.) was used, for quantification ethyl *n*-decanoate as internal standard (Purity ≥ 98.0%, Sigma Aldrich, St. Louis, MO, USA.) was used. The same HS-SPME method was used for quantification by internal standard and for identification by alkanes.

### 3.7. Quality Assurance and Control

An analytical blank procedure, an internal reference standard, and a triplicate of each of the samples were routinely analyzed with each sample, respectively. In the analytical blank procedures, no detectable concentrations of any target analytes were found. The accuracy and precision were determined from the analysis of samples in triplicate and the authentic standard, were within the certified ranges and varied within the acceptable limits (relative percentage difference less than ± 25%). Assays of recovery of the authentic standards varied as follows: methyl acetate 87–93% methyl isovalerate 69–93%; isopropyl butyrate 65–68%; hexanoic acid methyl ester 83–91%; 3-carene 69–79%; limonene 73–85%; 1-hexanol, 2-ethyl 78–83%; linalool 82–91; nonanal 67–78%; nonadecane 85–93%; tetradecane 67–74%; α-farnesene 78–83%; α-muurolene 85–88%. The limits of detection (LOD) of volatile organic compounds were estimated as the lowest concentration of a surrogate standard present in the sample, providing an instrumental response equal to the signal observed with the lowest calibration solution. The LODs ranged from 0.05 to 0.64 ng/g.

### 3.8. Statistical Analysis

The significance of variation in the levels of 40 different volatiles across the seven maturation stage in the three varieties of strawberry was analyzed by univariate and multivariate statistics using RStudio software (version 1.0.143, 2009–2016 R.Studio, Inc., Boston, MA, USA) running the R version 4.0.2 (22 June 2020). The multiple mean comparison (*p*-value ≤ 0.05) analysis was carried out by using one-way analysis of variance (ANOVA) with Tukey’s Honest Significant Difference (HSD) test. Multivariate analysis was performed by exporting GC-MS data in Excel format to R.Studio software, MetaboAnalystR 4.0 package was used to make heatmaps (HM) using two hierarchical groupings, columns indicated the maturation stages of the strawberry fruits, and rows indicated the VOCs identified. Variable importance in projection (VIP) graph was generated by MetaboAnalystR 4.0 package, for each of the strawberry varieties; this graph shows the 15 most important VOCs.

## 4. Conclusions

We identified possible volatile organic markers associated with the aroma of strawberry fruits of the Albion, Festival, and Frontera varieties and documented their behavior during maturation. Here, we show evidence that the dynamic levels of butanoic acid methyl ester; hexanoic acid methyl ester; octylcyclohexane; cyclohexane, 1,1,2-trimethyl; Linalool; tetradecane and α-muurolene could be volatile markers for harvesting strawberry fruits acceptable to consumers. 

## Figures and Tables

**Figure 1 molecules-26-00504-f001:**
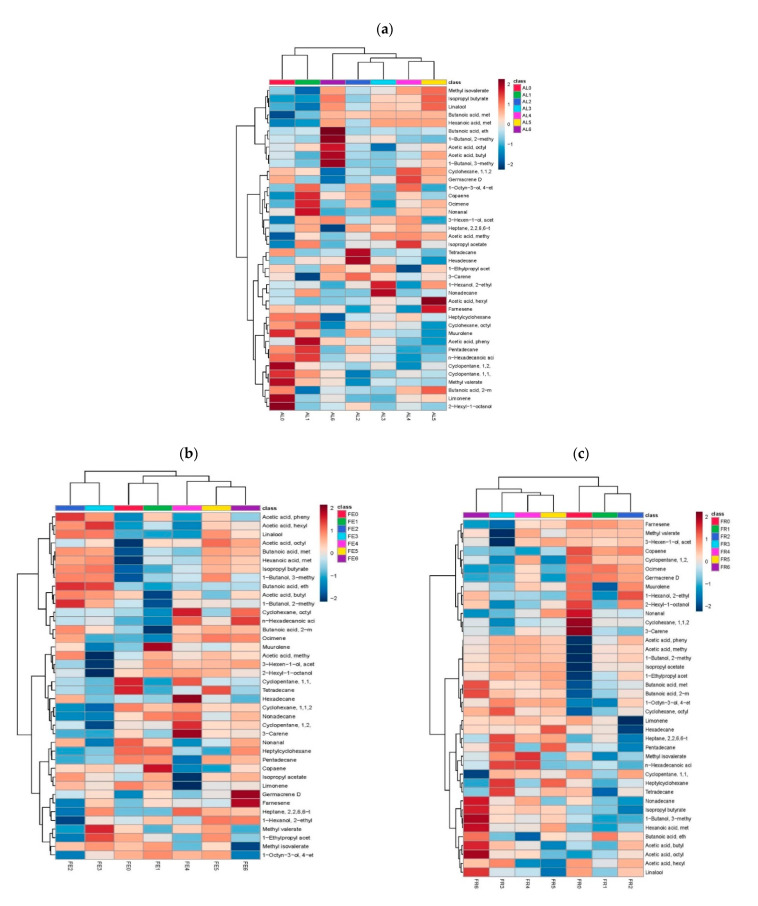
The heat map and hierarchical clustering analysis of metabolic profiles from maturation stages of the strawberry fruits. Rows: Volatile organic compounds (VOCs) identified; columns: maturation stages. The correlation matrix is represented by a color gradient in which dark blue indicates a negative correlation and dark red indicates positive correlation (Pearson Correlation Coefficient). This color matrix results in the grouping tree shown at the top of the figure, which groups the phenological maturation stages that do not differ significantly from each other on the same branch, and separates stages that do differ significantly on different branches; (**a**) heat map for var. Albion; (**b**) heat map for var. Festival; (**c**) heat map for var. Frontera.

**Figure 2 molecules-26-00504-f002:**
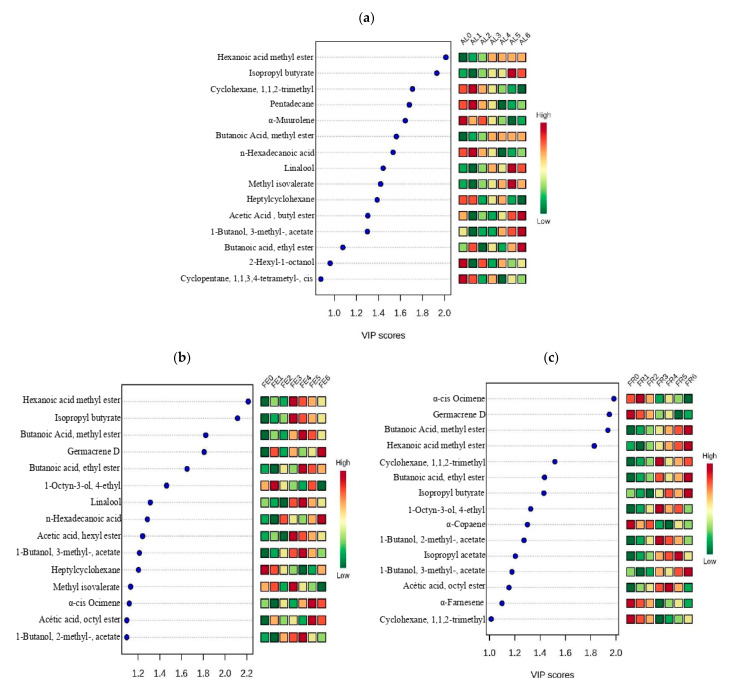
Variable importance in projection (VIP) graph showing the 15 most important VOCs in descending order. The degree of importance during each phenological stage is shown by the color gradient, in which red indicates a high correlation and green a low correlation. VIP score is a measure of a variable’s importance. (**a**) VIP var. Albion; (**b**) VIP var. Festival; (**c**) VIP var. Frontera.

**Figure 3 molecules-26-00504-f003:**
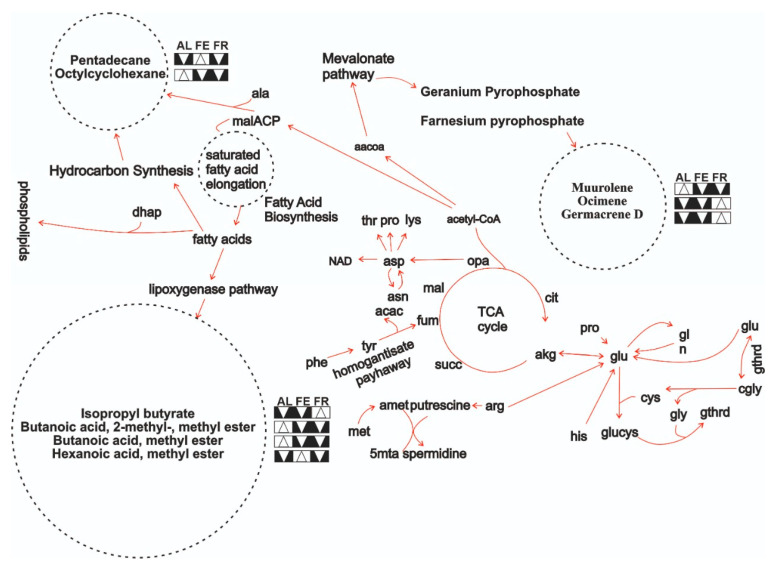
Metabolic pathways for the synthesis of the main VOCs detected in the Variable importance in projection (VIP) graph for fruits of the three strawberry varieties during the seven stages of maturation. The nine most important metabolites are classified into three chemical families; Compounds with white triangles facing upwards represent the highest total concentrations for the variety indicated on the labels, on the other hand, compounds with a black triangle facing downwards represent a lower total concentration of the metabolite for the variety indicated on the labels.

**Table 1 molecules-26-00504-t001:** Volatile organic compounds related to aroma in Albion variety strawberries at seven stages of maturation.

#	Compound	KI	AL0	AL1	AL2	AL3	AL4	AL5	AL6
1	Methyl acetate ^+^	498	0.22 ± 0.01 ^c^	3.60 ± 0.10 ^b^	5.68 ± 0.72 ^a^	5.31 ± 0.73 ^a^	5.38 ± 0.30 ^a^	3.94 ± 0.07 ^b^	5.72 ± 0.18 ^a^
2	Isopropyl acetate *	575	0.24 ± 0.03 ^f^	0.87 ± 0.06 ^e^	1.45 ± 0.21 ^dc^	1.77 ± 0.09 ^a^	1.55 ± 0.02 ^bc^	1.26 ± 0.12 ^d^	1.68 ± 0.05 ^ba^
3	Butanoic Acid, methyl ester *	724	0.08 ± 0.01 ^f^	2.26 ± 0.12 ^f^	8.85 ± 0.28 ^e^	26.79 ± 2.01 ^c^	18.81 ± 0.94 ^d^	33.58 ± 3.43 ^b^	42.42 ± 2.29 ^a^
4	Butanoic Acid, 2-methyl-, methyl ester *	783	0.22 ± 0.01 ^d^	0.01 ± 0.00 ^g^	0.09 ± 0.01 ^f^	0.11 ± 0.01 ^e^	0.25 ± 0.00 ^c^	0.39 ± 0.01 ^a^	0.33 ± 0.00 ^b^
5	Methyl isovalerate ^+^	783	0.15 ± 0.01 ^d^	0.01 ± 0.00 ^d^	0.33 ± 0.02 ^d^	0.97 ± 0.09 ^c^	1.19 ± 0.08 ^cb^	1.49 ± 0.22 ^b^	3.11 ± 0.40 ^a^
6	Butanoic acid, ethyl ester *	810	0.13 ± 0.01 ^b^	0.13 ± 0.02 ^b^	0.10 ± 0.01 ^b^	0.26 ± 0.02 ^b^	0.17 ± 0.00 ^b^	0.31 ± 0.00 ^b^	22.10 ± 2.72 ^a^
7	Cyclopentane, 1,1,3,4-tetrametyl-, cis *	817	0.64 ± 0.01 ^b^	0.25 ± 0.03 ^d^	0.02 ± 0.00 ^f^	0.15 ± 0.02 ^e^	0.57 ± 0.03 ^c^	0.27 ± 0.03 ^d^	0.88 ± 0.04 ^a^
8	Acetic Acid, butyl ester *	825	0.22 ± 0.01 ^c^	0.03 ± 0.00 ^c^	0.25 ± 0.01 ^c^	0.16 ± 0.01 ^c^	0.30 ± 0.02 ^c^	1.34 ± 0.06 ^b^	13.12 ± 0.62 ^a^
9	Methyl valerate *	835	0.23 ± 0.01 ^b^	0.12 ± 0.02 ^e^	0.05 ± 0.01 ^f^	0.17 ± 0.02 ^c^	0.15 ± 0.02 ^dc^	0.14 ± 0.02 ^de^	0.44 ± 0.04 ^a^
10	Isopropyl butyrate ^+^	854	0.09 ± 0.00 ^c^	0.04 ± 0.00 ^d^	0.37 ± 0.02 ^dc^	2.59 ± 0.12 ^c^	1.89 ± 0.02 ^dc^	7.78 ± 0.24 ^b^	20.57 ± 2.39 ^a^
11	1-Ethylpropyl acetate *	858	0.19 ± 0.01 ^d^	0.07 ± 0.00 ^e^	0.25 ± 0.03 ^dc^	0.44 ± 0.04 ^b^	0.01 ± 0.00 ^e^	0.32 ± 0.02 ^c^	0.76 ± 0.10 ^a^
12	Cyclopentane, 1,2,3,4,5-pentamethyl *	876	0.55 ± 0.03 ^a^	0.13 ± 0.02 ^d^	0.08 ± 0.01 ^e^	0.29 ± 0.02 ^c^	0.01 ± 0.00 ^f^	0.26 ± 0.02 ^c^	0.42 ± 0.04 ^b^
13	1-Butanol, 3-methyl-, acetate *	886	0.23 ± 0.02 ^de^	0.14 ± 0.02 ^e^	0.32 ± 0.04 ^dc^	0.42 ± 0.00 ^c^	0.88 ± 0.10 ^b^	0.77 ± 0.07 ^b^	4.59 ± 0.13 ^a^
14	1-Butanol, 2-methyl-, acetate *	888	0.10 ± 0.01 ^b^	0.02 ± 0.00 ^b^	0.21 ± 0.03 ^b^	0.33 ± 0.05 ^b^	0.01 ± 0.00 ^b^	0.02 ± 0.00 ^b^	6.62 ± 0.85 ^a^
15	Cyclohexane, 1,1,2-trimethyl *	889	4.37 ± 0.53 ^a^	1.71 ± 0.12 ^c^	1.72 ± 0.20 ^c^	1.29 ± 0.03 ^c^	4.01 ± 0.38 ^a^	2.50 ± 0.24 ^b^	0.26 ± 0.02 ^d^
16	Hexanoic acid methyl ester ^+^	933	0.04 ± 0.00 ^d^	0.10 ± 0.00 ^d^	2.06 ± 0.19 ^d^	10.06 ± 0.55 ^c^	7.31 ± 0.29 ^c^	17.23 ± 1.69 ^b^	32.79 ± 3.74 ^a^
17	Acetic acid, hexyl ester *	987	0.21 ± 0.01 ^dc^	0.10 ± 0.01 ^d^	0.28 ± 0.03 ^c^	0.60 ± 0.03 ^b^	0.51 ± 0.05 ^b^	1.40 ± 0.16 ^a^	0.49 ± 0.07 ^b^
18	Heptane, 2,2,6,6-tetramethyl *	1003	0.08 ± 0.00 ^c^	0.11 ± 0.01 ^c^	0.25 ± 0.01 ^a^	0.19 ± 0.02 ^b^	0.22 ± 0.03 ^a^	0.18 ± 0.01 ^b^	0.02 ± 0.00 ^d^
19	3-Hexen-1-ol, acetate, (*E*)- *	1013	0.01 ± 0.00 ^e^	0.10 ± 0.01 ^c^	0.07 ± 0.01 ^c^	0.20 ± 0.02 ^b^	0.20 ± 0.01 ^b^	0.03 ± 0.00 ^de^	0.61 ± 0.05 ^a^
20	3-Carene ^+^	1034	0.15 ± 0.01 ^d^	0.01 ± 0.00 ^e^	0.36 ± 0.02 ^b^	0.30 ± 0.02 ^c^	0.17 ± 0.03 ^d^	0.27 ± 0.02 ^c^	0.73 ± 0.06 ^a^
21	Limonene ^+^	1038	0.70 ± 0.05 ^a^	0.08 ± 0.00 ^b^	0.07 ± 0.01 ^b^	0.11 ± 0.00 ^b^	0.74 ± 0.06 ^a^	0.71 ± 0.09 ^a^	0.77 ± 0.10 ^a^
22	1-Hexanol, 2-ethyl ^+^	1040	0.11 ± 0.01 ^e^	0.09 ± 0.00 ^e^	0.17 ± 0.00 ^c^	0.76 ± 0.02 ^a^	0.07 ± 0.01 ^f^	0.35 ± 0.00 ^b^	0.21 ± 0.00 ^c^
23	α-cis Ocimene *	1052	0.02 ± 0.00 ^d^	0.17 ± 0.01 ^b^	0.14 ± 0.00 ^c^	0.02 ± 0.00 ^d^	0.14 ± 0.02 ^c^	0.22 ± 0.02 ^a^	0.03 ± 0.00 ^d^
24	1-Octyn-3-ol, 4-ethyl *	1095	0.06 ± 0.01 ^e^	0.12 ± 0.00 ^d^	0.23 ± 0.02 ^b^	0.10 ± 0.02 ^d^	0.26 ± 0.01 ^a^	0.07 ± 0.00 ^e^	0.20 ± 0.01 ^c^
25	Nonanal ^+^	1105	0.09 ± 0.01 ^d^	0.23 ± 0.01 ^b^	0.04 ± 0.01 ^e^	0.06 ± 0.01 ^e^	0.20 ± 0.01 ^c^	0.25 ± 0.00 ^a^	0.02 ± 0.00 ^f^
26	Linalool ^+^	1110	0.18 ± 0.00 ^d^	0.02 ± 0.00 ^d^	0.49 ± 0.05 ^d^	2.25 ± 0.15 ^b^	1.21 ± 0.07 ^c^	2.38 ± 0.10 ^b^	6.31 ± 0.48 ^a^
27	2-Hexyl-1-octanol *	1110	0.47 ± 0.04 ^a^	0.03 ± 0.00 ^f^	0.29 ± 0.03 ^c^	0.11 ± 0.02 ^e^	0.18 ± 0.01 ^d^	0.11 ± 0.01 ^e^	0.39 ± 0.01 ^b^
28	Acetic acid, phenylmethyl ester *	1169	0.12 ± 0.01 ^de^	0.18 ± 0.02 ^c^	0.15 ± 0.00 ^dc^	0.25 ± 0.04 ^b^	0.15 ± 0.01 ^dc^	0.08 ± 0.01 ^e^	0.57 ± 0.04 ^c^
29	Acétic acid, octyl ester *	1213	0.16 ± 0.02 ^cb^	0.14 ± 0.02 ^c^	0.17 ± 0.02 ^cb^	0.07 ± 0.01 ^c^	0.24 ± 0.00 ^cb^	0.38 ± 0.04 ^b^	2.25 ± 0.30 ^a^
30	α-Copaene *	1357	0.04 ± 0.00 ^d^	0.17 ± 0.01 ^c^	0.26 ± 0.03 ^b^	0.08 ± 0.01 ^d^	0.19 ± 0.02 ^c^	0.08 ± 0.00 ^d^	0.39 ± 0.06 ^a^
31	Heptylcyclohexane *	1359	7.30 ± 0.93 ^a^	5.62 ± 0.83 ^b^	4.61 ± 0.30 ^b^	2.69 ± 0.01 ^c^	2.18 ± 0.05 ^c^	2.24 ± 0.20 ^c^	0.62 ± 0.09 ^d^
32	Nonadecane ^+^	1407	0.15 ± 0.01 ^c^	1.61 ± 0.04 ^b^	0.05 ± 0.00 ^c^	3.84 ± 0.28 ^a^	0.02 ± 0.00 ^c^	0.29 ± 0.02 ^c^	0.16 ± 0.01 ^c^
33	Tetradecane ^+^	1415	5.50 ± 0.72 ^b^	0.10 ± 0.01 ^c^	8.70 ± 1.02 ^a^	0.31 ± 0.01 ^c^	0.95 ± 0.05 ^c^	0.12 ± 0.02 ^c^	1.03 ± 0.07 ^c^
34	Hexadecane *	1416	0.05 ± 0.00 ^c^	0.12 ± 0.02 ^c^	1.86 ± 0.28 ^a^	0.27 ± 0.01 ^c^	0.15 ± 0.01 ^c^	0.01 ± 0.00 ^c^	0.58 ± 0.07 ^b^
35	Octylcyclohexane *	1474	8.39 ± 0.45 ^ba^	9.15 ± 1.32 ^a^	7.31 ± 0.61 ^b^	2.93 ± 0.09 ^c^	1.89 ± 0.14 ^c^	0.18 ± 0.00 ^c^	0.18 ± 0.03 ^c^
36	Germacrene D *	1501	0.21 ± 0.03 ^dc^	0.06 ± 0.01 ^e^	0.14 ± 0.01 ^de^	0.25 ± 0.01 ^c^	0.88 ± 0.08 ^a^	0.34 ± 0.03 ^b^	0.12 ± 0.00 ^e^
37	α-Farnesene ^+^	1506	0.11 ± 0.01 ^c^	0.05 ± 0.01 ^d^	0.03 ± 0.00 ^d^	0.14 ± 0.00 ^c^	0.01 ± 0.00 ^d^	0.39 ± 0.04 ^a^	0.24 ± 0.03 ^b^
38	α-Muurolene ^+^	1507	12.96 ± 0.30 ^a^	4.83 ± 0.27 ^c^	7.99 ± 1.02 ^b^	2.01 ± 0.14 ^d^	1.00 ± 0.07 ^ed^	0.07 ± 0.01 ^e^	0.64 ± 0.01 ^e^
39	Pentadecane *	1605	6.23 ± 0.77 ^a^	6.64 ± 0.53 ^a^	5.84 ± 0.02 ^a^	2.22 ± 0.06 ^b^	0.19 ± 0.02 ^c^	0.35 ± 0.00 ^c^	0.87 ± 0.12 ^c^
40	*n*-Hexadecanoic acid *	1825	0.32 ± 0.02 ^d^	0.43 ± 0.01 ^b^	0.35 ± 0.04 ^dc^	0.38 ± 0.04 ^c^	0.12 ± 0.01 ^f^	0.24 ± 0.03 ^e^	0.50 ± 0.03 ^a^

The means ± standard deviations are presented. Different letters in superscript represent a significant difference between strawberry maturation stages according to Tukey test (*p* ≤ 0.05, *n* = 9); Unit: concentration (μL/100 g of strawberry samples, equivalent to ethyl *n*-decanoate); maturation stages in variety Albion: AL0, AL1, AL2, AL3, AL4, AL5 and AL6; KI: Kovats index; * tentatively identified by mass spectrometry and KI; ^+^ compounds identified using authentic standard; The KI values correspond to data reported previously in the literature or in the database on the website (http://www.nist.gov).

**Table 2 molecules-26-00504-t002:** Volatile organic compounds related to aroma in Festival variety strawberries at seven stages of maturation.

#	Compound	KI	FE0	FE1	FE2	FE3	FE4	FE5	FE6
1	Methyl acetate ^+^	498	0.69 ± 0.08 ^d^	1.24 ± 0.08 ^dc^	1.73 ± 0.17 ^c^	4.77 ± 0.05 ^b^	5.86 ± 0.52 ^a^	4.64 ± 0.61 ^b^	1.81 ± 0.22 ^c^
2	Isopropyl acetate *	575	0.05 ± 0.00 ^e^	0.01 ± 0.00 ^e^	0.27 ± 0.03 ^d^	0.89 ± 0.07 ^a^	0.94 ± 0.06 ^a^	0.52 ± 0.05 ^c^	0.70 ± 0.03 ^b^
3	Butanoic Acid, methyl ester *	724	0.08 ± 0.00 ^d^	0.67 ± 0.05 ^d^	1.25 ± 0.06 ^d^	16.25 ± 0.91 ^c^	21.24 ± 1.19 ^b^	18.65 ± 0.08 ^cb^	32.61 ± 3.85 ^a^
4	Butanoic Acid, 2-methyl-, methyl ester *	783	0.02 ± 0.00 ^e^	0.10 ± 0.00 ^d^	0.00 ± 0.00 ^e^	0.51 ± 0.00 ^b^	0.82 ± 0.04 ^a^	0.54 ± 0.04 ^b^	0.40 ± 0.03 ^c^
5	Methyl isovalerate ^+^	783	0.06 ± 0.00 ^ed^	0.01 ± 0.00 ^e^	0.13 ± 0.02 ^cd^	0.01 ± 0.00 ^e^	0.24 ± 0.02 ^b^	0.18 ± 0.00 ^cb^	0.74 ± 0.09 ^a^
6	Butanoic acid, ethyl ester *	810	0.07 ± 0.01 ^c^	0.11 ± 0.01 ^c^	0.04 ± 0.00 ^c^	0.46 ± 0.06 ^c^	9.85 ± 0.36 ^b^	0.38 ± 0.02 ^c^	14.64 ± 2.01 ^a^
7	Cyclopentane, 1,1,3,4-tetrametyl-, cis *	817	0.38 ± 0.03 ^a^	0.42 ± 0.05 ^a^	0.07 ± 0.01 ^c^	0.42 ± 0.02 ^a^	0.33 ± 0.05 ^b^	0.38 ± 0.00 ^a^	0.42 ± 0.02 ^a^
8	Acetic Acid, butyl ester *	825	0.07 ± 0.01 ^d^	0.09 ± 0.01 ^d^	0.00 ± 0.00 ^d^	0.31 ± 0.04 ^c^	0.75 ± 0.11 ^b^	0.05 ± 0.00 ^d^	1.06 ± 0.09 ^a^
9	Methyl valerate *	835	0.06 ± 0.01 ^dc^	0.09 ± 0.01 ^dc^	0.03 ± 0.01 ^d^	0.11 ± 0.01 ^c^	0.08 ± 0.01 ^dc^	0.35 ± 0.03 ^b^	1.27 ± 0.07 ^a^
10	Isopropyl butyrate ^+^	854	0.00 ± 0.00 ^d^	0.13 ± 0.01 ^d^	0.02 ± 0.00 ^d^	1.67 ± 0.07 ^c^	4.65 ± 0.49 ^b^	2.33 ± 0.20 ^c^	11.95 ± 0.63 ^a^
11	1-Ethylpropyl acetate *	858	0.06 ± 0.01 ^d^	0.03 ± 0.00 ^d^	0.05 ± 0.01 ^d^	0.11 ± 0.01 ^c^	0.05 ± 0.00 ^d^	0.28 ± 0.03 ^b^	0.73 ± 0.03 ^a^
12	Cyclopentane, 1,2,3,4,5-pentamethyl *	876	0.07 ± 0.00 ^e^	0.32 ± 0.01 ^b^	0.10 ± 0.01 ^ed^	0.49 ± 0.05 ^a^	0.26 ± 0.02 ^c^	0.36 ± 0.00 ^b^	0.14 ± 0.01 ^d^
13	1-Butanol, 3-methyl-, acetate *	886	0.01 ± 0.00 ^d^	0.04 ± 0.00 ^d^	0.03 ± 0.00 ^d^	0.13 ± 0.01 ^c^	0.42 ± 0.06 ^b^	0.39 ± 0.01 ^b^	0.71 ± 0.02 ^a^
14	1-Butanol, 2-methyl-, acetate *	888	0.05 ± 0.00 ^c^	0.13 ± 0.00 ^a^	0.00 ± 0.00 ^cb^	0.35 ± 0.03 ^cb^	7.74 ± 0.46 ^cb^	0.39 ± 0.02 ^cb^	2.30 ± 0.01 ^cb^
15	Cyclohexane, 1,1,2-trimethyl *	889	2.58 ± 0.10 ^c^	3.09 ± 0.42 ^c^	1.75 ± 0.13 ^d^	5.07 ± 0.73 ^a^	0.38 ± 0.04 ^e^	3.85 ± 0.07 ^b^	0.34 ± 0.03 ^e^
16	Hexanoic acid methyl ester ^+^	933	0.00 ± 0.00 ^d^	0.42 ± 0.05 ^d^	0.83 ± 0.04 ^d^	12.40 ± 0.55 ^c^	24.01 ± 2.62 ^b^	21.24 ± 2.63 ^b^	66.02 ± 6.34 ^a^
17	Acetic acid, hexyl ester *	987	0.02 ± 0.00 ^c^	0.02 ± 0.00 ^c^	0.09 ± 0.01 ^c^	0.59 ± 0.05 ^c^	1.81 ± 0.04 ^b^	0.78 ± 0.08 ^cb^	8.96 ± 1.13 ^a^
18	Heptane, 2,2,6,6-tetramethyl *	1003	0.01 ± 0.00 ^d^	0.13 ± 0.02 ^c^	0.02 ± 0.00 ^d^	0.22 ± 0.03 ^b^	0.03 ± 0.00 ^d^	0.22 ± 0.02 ^b^	0.52 ± 0.05 ^a^
19	3-Hexen-1-ol, acetate, (*E*)- *	1013	0.04 ± 0.00 ^d^	0.09 ± 0.00 ^c^	0.14 ± 0.00 ^c^	0.45 ± 0.02 ^a^	0.11 ± 0.01 ^c^	0.31 ± 0.05 ^b^	0.05 ± 0.00 ^d^
20	3-Carene ^+^	1034	0.09 ± 0.01 ^dc^	1.32 ± 0.02 ^a^	0.04 ± 0.00 ^e^	0.35 ± 0.02 ^b^	0.07 ± 0.00 ^d^	0.33 ± 0.01 ^b^	0.11 ± 0.00 ^c^
21	Limonene ^+^	1038	0.42 ± 0.01 ^c^	0.00 ± 0.00 ^e^	0.33 ± 0.04 ^d^	0.44 ± 0.02 ^c^	0.88 ± 0.04 ^a^	0.31 ± 0.03 ^d^	0.70 ± 0.09 ^b^
22	1-Hexanol, 2-ethyl ^+^	1040	0.02 ± 0.00 ^c^	0.03 ± 0.00 ^c^	0.07 ± 0.01 ^c^	0.44 ± 0.06 ^a^	0.01 ± 0.00 ^c^	0.39 ± 0.02 ^a^	0.23 ± 0.01 ^b^
23	α-cis Ocimene *	1052	0.03 ± 0.00 ^f^	0.10 ± 0.00 ^e^	0.03 ± 0.00 ^f^	0.43 ± 0.02 ^a^	0.32 ± 0.03 ^c^	0.38 ± 0.04 ^b^	0.20 ± 0.01 ^d^
24	1-Octyn-3-ol, 4-ethyl *	1095	0.05 ± 0.00 ^d^	0.07 ± 0.01 ^d^	0.13 ± 0.02 ^c^	0.01 ± 0.00 ^e^	0.01 ± 0.00 ^e^	0.29 ± 0.03 ^a^	0.17 ± 0.01 ^b^
25	Nonanal ^+^	1105	0.09 ± 0.00 ^ed^	0.05 ± 0.00 ^e^	0.10 ± 0.01 ^d^	0.39 ± 0.04 ^a^	0.32 ± 0.03 ^b^	0.23 ± 0.00 ^c^	0.25 ± 0.00 ^c^
26	Linalool ^+^	1110	0.09 ± 0.00 ^e^	0.11 ± 0.01 ^e^	0.12 ± 0.01 ^e^	1.59 ± 0.23 ^d^	3.48 ± 0.20 ^b^	2.65 ± 0.18 ^c^	5.79 ± 0.06 ^a^
27	2-Hexyl-1-octanol *	1110	0.05 ± 0.00 ^d^	0.14 ± 0.02 ^c^	0.14 ± 0.01 ^c^	0.22 ± 0.03 ^b^	0.14 ± 0.02 ^c^	0.36 ± 0.04 ^a^	0.02 ± 0.00 ^d^
28	Acetic acid, phenylmethyl ester *	1169	0.04 ± 0.01 ^d^	0.06 ± 0.01 ^d^	0.13 ± 0.00 ^d^	0.27 ± 0.01 ^c^	0.65 ± 0.09 ^b^	0.35 ± 0.03 ^c^	0.83 ± 0.10 ^a^
29	Acétic acid, octyl ester *	1213	0.01 ± 0.00 ^e^	0.05 ± 0.00 ^d^	0.05 ± 0.00 ^d^	0.25 ± 0.00 ^b^	0.11 ± 0.01 ^c^	0.27 ± 0.02 ^a^	0.26 ± 0.02 ^ba^
30	α-Copaene *	1357	0.02 ± 0.00 ^d^	0.00 ± 0.00 ^d^	3.06 ± 0.12 ^a^	0.24 ± 0.02 ^c^	0.24 ± 0.03 ^c^	0.01 ± 0.00 ^d^	0.57 ± 0.06 ^b^
31	Heptylcyclohexane *	1359	3.07 ± 0.46 ^c^	0.08 ± 0.00 ^d^	4.52 ± 0.22 ^b^	8.59 ± 0.47 ^a^	0.05 ± 0.01 ^d^	0.06 ± 0.01 ^d^	0.24 ± 0.03 ^d^
32	Nonadecane ^+^	1407	1.19 ± 0.02 ^e^	11.27 ± 0.92 ^b^	8.08 ± 0.10 ^c^	12.43 ± 0.77 ^a^	0.25 ± 0.01 ^e^	2.57 ± 0.26 ^d^	0.35 ± 0.02 ^e^
33	Tetradecane ^+^	1415	6.85 ± 0.32 ^b^	0.14 ± 0.01 ^c^	0.02 ± 0.00 ^c^	0.23 ± 0.02 ^c^	0.24 ± 0.01 ^c^	14.36 ± 0.52 ^a^	0.16 ± 0.02 ^c^
34	Hexadecane *	1416	0.07 ± 0.00 ^e^	1.76 ± 0.06 ^a^	0.08 ± 0.01 ^e^	0.19 ± 0.01 ^d^	0.98 ± 0.06 ^b^	0.39 ± 0.03 ^c^	0.34 ± 0.02 ^c^
35	Octylcyclohexane *	1474	0.06 ± 0.01 ^c^	7.10 ± 0.72 ^b^	0.01 ± 0.00 ^c^	8.68 ± 0.20 ^a^	0.35 ± 0.02 ^c^	0.20 ± 0.01 ^c^	0.75 ± 0.04 ^c^
36	Germacrene D *	1501	0.00 ± 0.00 ^e^	0.02 ± 0.00 ^de^	0.10 ± 0.01 ^dc^	9.51 ± 0.09 ^a^	0.12 ± 0.00 ^c^	0.18 ± 0.02 ^c^	0.47 ± 0.00 ^b^
37	α-Farnesene ^+^	1506	0.01 ± 0.00 ^c^	0.06 ± 0.01 ^c^	0.09 ± 0.01 ^c^	7.10 ± 0.25 ^a^	0.01 ± 0.00 ^c^	0.13 ± 0.01 ^c^	0.69 ± 0.04 ^b^
38	α-Muurolene ^+^	1507	0.02 ± 0.00 ^c^	0.07 ± 0.01 ^c^	2.60 ± 0.18 ^a^	0.43 ± 0.06 ^b^	0.34 ± 0.00 ^b^	0.16 ± 0.01 ^c^	0.03 ± 0.00 ^c^
39	Pentadecane *	1605	3.84 ± 0.22 ^c^	0.01 ± 0.00 ^e^	4.08 ± 0.12 ^c^	6.82 ± 0.15 ^b^	0.12 ± 0.01 ^e^	8.64 ± 0.12 ^a^	0.56 ± 0.01 ^d^
40	*n*-Hexadecanoic acid *	1825	0.02 ± 0.00 ^d^	0.47 ± 0.02 ^b^	0.02 ± 0.00 ^d^	2.31 ± 0.19 ^a^	0.21 ± 0.03 ^c^	0.35 ± 0.03 ^cb^	0.50 ± 0.05 ^b^

The means ± standard deviations are presented. Different letters in superscript represent a significant difference between strawberry maturation stages according to Tukey test (*p* ≤ 0.05, *n* = 9); Unit: concentration (μL/100 g of strawberry samples, equivalent to ethyl *n*-decanoate); maturation stages in variety Festival: FE0, FE1, FE2, FE3, FE4, FE5, and FE6; KI: Kovats index; * tentatively identified by mass spectrometry and KI; ^+^ compounds identified using authentic standard; The KI values correspond to data reported previously in the literature or in the database on the website (http://www.nist.gov).

**Table 3 molecules-26-00504-t003:** Volatile organic compounds related to aroma in Frontera variety strawberries at seven stages of maturation.

#	Compound	KI	FR0	FR1	FR2	FR3	FR4	FR5	FR6
1	Methyl acetate ^+^	498	2.13 ± 0.22 ^c^	81.61 ± 4.00 ^a^	4.00 ± 0.02 ^c^	3.81 ± 0.31 ^c^	4.82 ± 0.03 ^cb^	5.97 ± 0.76 ^cb^	8.15 ± 0.25 ^b^
2	Isopropyl acetate *	575	2.50 ± 0.20 ^d^	42.07 ± 0.67 ^a^	1.68 ± 0.02 ^ef^	1.60 ± 0.18 ^f^	2.24 ± 0.23 ^ed^	3.99 ± 0.05 ^c^	5.90 ± 0.26 ^b^
3	Butanoic Acid, methyl ester *	724	0.46 ± 0.04 ^d^	7.37 ± 0.54 ^b^	0.47 ± 0.02 ^d^	0.45 ± 0.03 ^d^	0.64 ± 0.08 ^d^	2.10 ± 0.23 ^c^	16.23 ± 1.12 ^a^
4	Butanoic Acid, 2-methyl-, methyl ester *	783	0.97 ± 0.06 ^c^	4.82 ± 0.29 ^a^	0.36 ± 0.04 ^d^	0.31 ± 0.04 ^d^	0.28 ± 0.04 ^d^	0.55 ± 0.03 ^dc^	3.95 ± 0.52 ^b^
5	Methyl isovalerate ^+^	783	1.01 ± 0.08 ^a^	0.27 ± 0.02 ^b^	0.02 ± 0.00 ^c^	0.04 ± 0.01 ^c^	0.09 ± 0.01 ^c^	0.02 ± 0.00 ^c^	0.08 ± 0.00 ^c^
6	Butanoic acid, ethyl ester *	810	0.85 ± 0.11 ^b^	3.14 ± 0.09 ^a^	0.06 ± 0.00 ^d^	0.01 ± 0.00 ^d^	0.01 ± 0.00 ^d^	0.07 ± 0.00 ^d^	0.36 ± 0.01 ^c^
7	Cyclopentane, 1,1,3,4-tetrametyl-, cis *	817	13.29 ± 0.45 ^a^	7.68 ± 0.89 ^b^	0.62 ± 0.01 ^c^	0.33 ± 0.02 ^c^	0.36 ± 0.04 ^c^	0.48 ± 0.05 ^c^	0.35 ± 0.04 ^c^
8	Acetic Acid, butyl ester *	825	1.25 ± 0.18 ^a^	0.97 ± 0.02 ^b^	0.14 ± 0.01 ^c^	0.08 ± 0.01 ^c^	0.06 ± 0.01 ^c^	0.03 ± 0.00 ^c^	1.00 ± 0.10 ^b^
9	Methyl valerate *	835	2.50 ± 0.34 ^d^	3.06 ± 0.44 ^d^	0.12 ± 0.01 ^d^	0.01 ± 0.00 ^c^	0.10 ± 0.01 ^b^	0.10 ± 0.01 ^d^	0.27 ± 0.01 ^a^
10	Isopropyl butyrate ^+^	854	0.85 ± 0.09 ^b^	0.85 ± 0.13 ^b^	0.01 ± 0.00 ^c^	0.06 ± 0.01 ^c^	0.10 ± 0.01 ^c^	0.14 ± 0.01 ^c^	1.82 ± 0.02 ^a^
11	1-Ethylpropyl acetate *	858	0.82 ± 0.03 ^b^	4.57 ± 0.39 ^a^	0.31 ± 0.02 ^cb^	0.24 ± 0.03 ^d^	0.24 ± 0.02 ^d^	0.50 ± 0.02 ^cbd^	0.67 ± 0.04 ^cb^
12	Cyclopentane, 1,2,3,4,5-pentamethyl *	876	4.61 ± 0.02 ^a^	2.52 ± 0.24 ^b^	0.10 ± 0.00 ^c^	0.01 ± 0.00 ^c^	0.09 ± 0.00 ^c^	0.02 ± 0.00 ^c^	0.13 ± 0.02 ^c^
13	1-Butanol, 3-methyl-, acetate *	886	1.74 ± 0.15 ^a^	1.09 ± 0.16 ^b^	0.05 ± 0.01 ^c^	0.07 ± 0.01 ^c^	0.07 ± 0.00 ^c^	0.18 ± 0.02 ^c^	1.61 ± 0.22 ^a^
14	1-Butanol, 2-methyl-, acetate *	888	1.28 ± 0.01 ^c^	5.12 ± 6.56 ^a^	5.71 ± 0.17 ^cb^	5.20 ± 0.61 ^cb^	5.48 ± 0.23 ^b^	7.89 ± 0.25 ^cb^	10.68 ± 0.24 ^b^
15	Cyclohexane, 1,1,2-trimethyl *	889	52.60 ± 6.26 ^a^	3.37 ± 0.15 ^b^	0.13 ± 0.02 ^b^	0.01 ± 0.00 ^b^	0.06 ± 0.01 ^b^	0.13 ± 0.01 ^b^	0.31 ± 0.04 ^b^
16	Hexanoic acid methyl ester ^+^	933	2.83 ± 0.08 ^b^	2.25 ± 0.21 ^b^	0.19 ± 0.01 ^c^	0.26 ± 0.01 ^c^	0.35 ± 0.05 ^c^	1.81 ± 0.19 ^cb^	21.03 ± 1.58 ^a^
17	Acetic acid, hexyl ester *	987	2.34 ± 0.11 ^a^	1.98 ± 0.26 ^b^	0.10 ± 0.01 ^c^	0.07 ± 0.01 ^c^	0.03 ± 0.00 ^c^	0.05 ± 0.01 ^c^	0.27 ± 0.01 ^c^
18	Heptane, 2,2,6,6-tetramethyl *	1003	1.74 ± 0.15 ^b^	5.01 ± 0.22 ^a^	0.05 ± 0.01 ^d^	0.37 ± 0.04 ^c^	0.22 ± 0.03 ^dc^	0.42 ± 0.03 ^c^	0.37 ± 0.04 ^c^
19	3-Hexen-1-ol, acetate, (*E*)- *	1013	8.64 ± 5.25 ^a^	5.65 ± 4.57 ^a^	2.39 ± 0.34 ^b^	0.02 ± 0.00 ^b^	1.51 ± 0.12 ^b^	4.42 ± 0.62 ^b^	2.87 ± 0.36 ^b^
20	3-Carene ^+^	1034	12.52 ± 0.90 ^a^	1.40 ± 0.08 ^b^	0.04 ± 0.00 ^c^	0.02 ± 0.00 ^c^	0.07 ± 0.01 ^c^	0.09 ± 0.01 ^c^	0.20 ± 0.01 ^c^
21	Limonene ^+^	1038	2.56 ± 0.28 ^a^	2.29 ± 0.32 ^a^	0.03 ± 0.00 ^b^	0.06 ± 0.01 ^b^	0.08 ± 0.01 ^b^	0.11 ± 0.01 ^b^	0.27 ± 0.03 ^b^
22	1-Hexanol, 2-ethyl ^+^	1040	31.11 ± 3.61 ^a^	0.35 ± 0.04 ^b^	2.19 ± 0.13 ^b^	0.04 ± 0.00 ^b^	0.05 ± 0.01 ^b^	0.05 ± 0.01 ^b^	1.31 ± 0.16 ^b^
23	α-cis Ocimene *	1052	14.43 ± 1.22 ^b^	21.56 ± 6.06 ^a^	3.72 ± 0.43 ^c^	0.05 ± 0.00 ^c^	0.85 ± 0.05 ^c^	0.12 ± 0.00 ^c^	0.16 ± 0.02 ^c^
24	1-Octyn-3-ol, 4-ethyl *	1095	0.21 ± 0.02 ^b^	0.27 ± 0.04 ^a^	0.14 ± 0.01 ^c^	0.20 ± 0.01 ^b^	0.15 ± 0.01 ^c^	0.28 ± 0.03 ^a^	0.26 ± 0.01 ^a^
25	Nonanal ^+^	1105	20.06 ± 1.71 ^a^	5.26 ± 0.25 ^b^	0.18 ± 0.02 ^c^	0.06 ± 0.00 ^c^	0.15 ± 0.01 ^c^	0.54 ± 0.04 ^c^	0.22 ± 0.03 ^c^
26	Linalool ^+^	1110	2.86 ± 0.41 ^a^	0.35 ± 0.01 ^c^	0.08 ± 0.01 ^c^	0.02 ± 0.00 ^c^	0.02 ± 0.00 ^c^	0.00 ± 0.00 ^c^	1.27 ± 0.10 ^b^
27	2-Hexyl-1-octanol *	1110	3.01 ± 0.23 ^a^	2.40 ± 0.34 ^b^	0.13 ± 0.01 ^c^	0.05 ± 0.00 ^c^	0.07 ± 0.00 ^c^	0.13 ± 0.01 ^c^	0.33 ± 0.03 ^c^
28	Acetic acid, phenylmethyl ester *	1169	1.98 ± 0.17 ^b^	13.25 ± 0.50 ^a^	0.85 ± 0.03 ^c^	0.53 ± 0.00 ^c^	0.66 ± 0.05 ^c^	0.92 ± 0.02 ^c^	1.56 ± 0.01 ^b^
29	Acétic acid, octyl ester *	1213	1.16 ± 0.00 ^b^	2.33 ± 0.25 ^a^	0.13 ± 0.01 ^c^	0.08 ± 0.01 ^c^	0.10 ± 0.01 ^c^	0.08 ± 0.00 ^c^	1.32 ± 0.18 ^b^
30	α-Copaene *	1357	12.80 ± 1.83 ^a^	8.89 ± 0.36 ^b^	0.42 ± 0.01 ^c^	0.04 ± 0.00 ^c^	0.04 ± 0.00 ^c^	0.12 ± 0.01 ^c^	0.21 ± 0.01 ^c^
31	Heptylcyclohexane *	1359	5.42 ± 0.79 ^a^	2.64 ± 0.25 ^b^	0.21 ± 0.00 ^c^	0.55 ± 0.03 ^c^	0.10 ± 0.01 ^c^	0.88 ± 0.13 ^c^	0.26 ± 0.01 ^c^
32	Nonadecane ^+^	1407	1.10 ± 0.11 ^a^	0.74 ± 0.02 ^b^	0.02 ± 0.00 ^d^	0.03 ± 0.00 ^d^	0.05 ± 0.00 ^d^	0.03 ± 0.00 ^d^	0.38 ± 0.03 ^c^
33	Tetradecane ^+^	1415	23.21 ± 1.44 ^a^	2.71 ± 0.33 ^b^	0.68 ± 0.07 ^c^	1.75 ± 0.18 ^cb^	0.40 ± 0.00 ^c^	1.36 ± 0.03 ^cb^	0.89 ± 0.01 ^c^
34	Hexadecane *	1416	2.41 ± 0.14 ^a^	2.68 ± 0.32 ^a^	0.01 ± 0.00 ^d^	0.07 ± 0.00 ^cd^	0.09 ± 0.01 ^cd^	0.55 ± 0.02 ^b^	0.38 ± 0.01 ^cb^
35	Octylcyclohexane *	1474	0.06 ± 0.00 ^fe^	0.50 ± 0.01 ^a^	0.07 ± 0.00 ^e^	0.33 ± 0.01 ^c^	0.05 ± 0.00 ^f^	0.16 ± 0.01 ^d^	0.49 ± 0.01 ^b^
36	Germacrene D *	1501	17.53 ± 3.71 ^a^	18.78 ± 6.03 ^a^	3.89 ± 0.37 ^b^	0.08 ± 0.01 ^b^	1.03 ± 0.11 ^b^	0.10 ± 0.01 ^b^	0.29 ± 0.00 ^b^
37	α-Farnesene ^+^	1506	45.02 ± 5.49 ^a^	51.11 ± 3.73 ^a^	1.36 ± 0.20 ^b^	0.03 ± 0.00 ^b^	0.53 ± 0.00 ^b^	1.08 ± 0.05 ^b^	0.31 ± 0.03 ^b^
38	α-Muurolene ^+^	1507	14.61 ± 0.66 ^a^	0.12 ± 0.01 ^b^	0.50 ± 0.07 ^b^	0.02 ± 0.00 ^b^	0.14 ± 0.01 ^b^	0.13 ± 0.01 ^b^	0.25 ± 0.03 ^b^
39	Pentadecane *	1605	2.44 ± 0.14 ^b^	3.65 ± 0.08 ^a^	0.03 ± 0.00 ^e^	0.72 ± 0.08 ^d^	0.05 ± 0.00 ^e^	1.25 ± 0.11 ^c^	0.62 ± 0.08 ^d^
40	*n*-Hexadecanoic acid *	1825	2.16 ± 0.03 ^b^	3.79 ± 0.55 ^a^	0.08 ± 0.01 ^c^	0.39 ± 0.02 ^c^	0.49 ± 0.07 ^c^	0.11 ± 0.01 ^c^	0.42 ± 0.03 ^c^

The means ± standard deviations are presented. Different letters in superscript represent a significant difference between strawberry maturation stages according to Tukey test (*p* ≤ 0.05, *n* = 9); Unit: concentration (μL/100 g of strawberry samples, equivalent to ethyl *n*-decanoate); maturation stages in variety Frontera: FR0, FR1, FR2, FR3, FR4, FR5, and FR6; KI: Kovats index; * tentatively identified by mass spectrometry and KI; ^+^ compounds identified using authentic standard; The KI values correspond to data reported previously in the literature or in the database on the website (http://www.nist.gov).

## Data Availability

The data presented in this study are available in this article.

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
