# Peer review of "Identification of Organic Volatile Markers Associated with Aroma during Maturation of Strawberry Fruits"

_molecules, 2021, doi:10.3390/molecules26020504_

Round 1
Reviewer 1 Report
Comments:
I have reviewed the manuscript entitled “Identification of Organic Volatile Markers Associated with Aroma During Maturation of Strawberry Fruits”.
This work was aimed to identify possible volatile organic compounds related to aroma
to monitor the maturation of strawberry fruits by using using SPME-GC-MS. This work is interesting and the result is helpful to monitor the maturation of strawberry fruits.
Nevertheless, some improvements would make it more clear.
Materials and Methods
- Line 99-108: The method of SPME has been widely to extract volatiles. However, the time of extraction is doubtful. Three minutes of extraction is too short to completely adsorb volatile compounds in this study. According to other studies, in order to extract the volatiles completely, more than half an hour of extraction time is essential (A, H. L. , A, K. W. , A, L. W. , A, D. L. , A, J. Y. , & B, Z. B. , et al. (2018)).
Results and discussion
- Figure 2: The compounds on the left of Figure 2 are not clearly labeled.
In addition, differences in volatile compounds as markers in different varieties should require appropriate discussion.
Conclusions
In this section, the findings in this study need to be more specific.
Reviewer 2 Report
The manuscript by Padilla-Jiménez et al. shows the comparison of the volatile organic compounds released from strawberries at different stages of their maturation. The study was well planned, and the authors provide good detail in their methodologies. I recommend publication after addressing minor comments.
The article is generally well written, and the discussion can be understood but there are multiple grammatical errors throughout the manuscript, usually to do with the tense of the sentence. Furthermore, the authors overuse first person, many ‘we’ statements, these should be changed to third person.
The samples were frozen and then analyzed by SPME. How were the samples thawed before the SPME?
In Tables 1-3 please provide the number of replicates in the caption and provide more details about the superscripted letters, from which stages are they significantly different?
In the Figure 1 caption, the description of the rows and columns are reversed, rows are horizontal, and columns are vertical.
The heat maps require more explanation for a general chemistry journal such as Molecules. What correlation do the colors show? What does a negative and positive correlation mean? Is it purely linked to changes in concentration?
I am surprised that the VOC profile is so different between the 3 strawberry varieties. Can a reason be given in the text as to why it is so different? Is this observed in other fruits?
Reviewer 3 Report
Manuscript: molecules-1051705
The manuscript present the evolution of the organic volatile of three strawberry varieties during maturation. The experimental design is simple but adequate to the objectives of the work. The presentation is clear and easy to follow. The following suggestions could be consider:
The name of the chemical compounds are all written in capital letters. This reviewer consider that it could be reasonable when they initiate a sentence but not within text.
-The statistical analysis should be rewritten. The RStudio is just a framework where R packages are run. Then, R version, packages used (stats, ggplot, etc) should be included. In addition, the variable importance graph, need some explanation and the package used for its performance should be included. It is important to provide sufficient information as to allow readers to reproduce it.
-Finally, the explanation of the possible metabolic pathways involved in the synthesis of the main VOCs is just outlined. This reviewer consider that such information deserves a more extensive explanation.
Regarding the English, several sentences could be expressed in shortest expressions.
Round 2
Reviewer 1 Report
I have carefully read through this revised manuscript. Overall the authors have responded to what I suggested although some answers were not satisfactory. In the present state, I think that their research findings can still have some value in strawberry research even though it is not perfect.I have no more questions about this work.
Author Response
Point 1: I have carefully read through this revised manuscript. Overall the authors have responded to what I suggested although some answers were not satisfactory. In the present state, I think that their research findings can still have some value in strawberry research even though it is not perfect. I have no more questions about this work.
Response 1: Dear reviewer, thank you for your time. Your comments and suggestions were well received. They certainly improved the work.
Reviewer 3 Report
Most of the suggestions were introduced in the revised versión. The only last suggestion reffers to the inclusion of the R versión used for the Statisitical analysis.
Author Response
Point 1: Most of the suggestions were introduced in the revised versión. The only last suggestion reffers to the inclusion of the R versión used for the Statisitical analysis.
Response 1: Dear reviewer, thank you for your time. Your comments and suggestions were well received. R version was included. They certainly improved the work.
Line: 134-135: “univariate and multivariate statistics using RStudio software (version 1.0.143, 2009-2016 R Studio, Inc.) running the R version 4.0.2 (2020-06-22).”